# Orthogonal translation enables heterologous ribosome engineering in *E. coli*

Natalie S. Kolber [1,2], Ranan Fattal[1], Sinisa Bratulic [1,3], Gavriela D. Carver [1] & Ahmed H. Badran [1✉]

The ribosome represents a promising avenue for synthetic biology, but its complexity and essentiality have hindered significant engineering efforts. Heterologous ribosomes, comprising rRNAs and r-proteins derived from different microorganisms, may offer opportunities for novel translational functions. Such heterologous ribosomes have previously been evaluated in *E. coli* via complementation of a genomic ribosome deficiency, but this method fails to guide the engineering of refractory ribosomes. Here, we implement orthogonal ribosome binding site (RBS):antiRBS pairs, in which engineered ribosomes are directed to researcher-defined transcripts, to inform requirements for heterologous ribosome functionality. We discover that optimized rRNA processing and supplementation with cognate r-proteins enhances heterologous ribosome function for rRNAs derived from organisms with ≥76.1% 16S rRNA identity to *E. coli*. Additionally, some heterologous ribosomes undergo reduced subunit exchange with *E. coli*-derived subunits. Cumulatively, this work provides a general framework for heterologous ribosome engineering in living cells.

[1] Broad Institute of MIT and Harvard, Cambridge, MA 02142, USA. [2]Present address: Department of Bioengineering, Stanford University, Stanford, CA 94305, USA. [3]Present address: Department of Biology and Biological Engineering, Chalmers University of Technology, Kemivägen 10, SE-412 96 Gothenburg, Sweden. ✉email: ahbadran@broadinstitute.org

I n addition to catalyzing the biosynthesis of the complete cellular proteome, the ribosome serves as a hub for signaling events, integrating nutrient availability with growth dynamics and resource allocation[1,2]. In prokaryotes, this functionality is enabled by the concerted action of numerous components: the 16S rRNA and 21 ribosomal proteins (r-proteins) define the small subunit (SSU or 30S), with the 23S rRNA, 5S rRNA, and 33 r-proteins defining the large subunit (LSU or 50S)[3]. Extensive efforts towards engineering translation have yielded researcher-dictated, specialized functions in vivo: parallel genetic circuits[4], augmented polypeptide diversity using non-canonical amino acids[5], expanded genetic codes incorporating quadruplet codons[6], and linked ribosomal subunits for improved cellular orthogonality[7–9]. However, these efforts have typically made use of solely E. coli components.

Conversely, ribosomal components derived from divergent microorganisms may offer the opportunity to discover unique or dedicated ribosomal capabilities, as suggested by naturally occurring subpopulations of prokaryotic ribosomes[10]. Indeed, stress-inducible production of rrsH ribosomes in E. coli modifies the cellular translational program[11], rrnI ribosomes in Vibrio vulnificius selectively translate certain mRNAs[12], and genetically heterogeneous ribosomes are produced at defined stages of the Streptomyces coelicolor developmental cycle[13]. R-protein complements may further specialize ribosomal function, exemplified by the hypothesized role of ribosomes carrying S1 in leaderless mRNA decoding[14] and the variation in r-protein complements during different growth conditions[15]. Heterologous ribosomes, synthesized in E. coli using rRNAs and r-proteins derived from divergent bacterial species, may therefore serve as foundations for the discovery or engineering of novel translational capabilities.

Prior investigations of heterologous rRNAs have been enabled by E. coli Δ7 strains lacking all seven chromosomal rRNA operons (e.g., SQ171, KT101, SQZ10, SQ2518)[16–18]. Such Δ7 strains have additionally informed studies on rDNA copy number[19], ribosomal sequence–function relationships[20], factors affecting rRNA processing[21,22], and rRNA–protein interactions[23]. These strains bear a complete genomic rRNA deficiency and are complemented by a counter-selectable rRNA-encoding plasmid, facilitating plasmid exchange with rRNA variants capable of sustaining E. coli survival. Promisingly, full-length heterologous rRNA operons derived from species bearing ≥93.2% 16S sequence identity to the E. coli counterpart have been found to sustain E. coli Δ7 strain viability[18], whereas 16S sequence fragments bearing ≥80.9% identity can substitute for otherwise wild-type E. coli 16S rRNAs[17]. Natural horizontal gene transfer events in the evolutionary record provide further evidence for heterologous translation with intragenomic 16S identity as low as 88.4%[24–28].

However, E. coli Δ7 strain complementation assays prove problematic for systematically evaluating heterologous ribosome function given the myriad roles played by the ribosome in sustaining cell viability, both catalytic and regulatory[29,30]. Efforts to engineer heterologous ribosome function require a more quantitative translational assay that reports on a single aspect of ribosome function. We therefore sought to develop guidelines for evaluating and enhancing the translational activity of heterologous rRNAs in E. coli using a method that reports exclusively on catalytic activity, independent of an rRNA's ability to support cell growth. To achieve this, we first constructed a library of 34 complete rRNA operons derived from phylogenetically diverse microbes. We evaluated their activities via E. coli Δ7 strain complementation as well as orthogonal translation. Here, engineered RBS:antiRBS pairs are leveraged to generate ribosomes that exclusively translate a researcher-defined transcript, which in turn cannot be translated by wildtype ribosomes[4,31–34]. Finding a high degree of correlation between the two methods, we applied

orthogonal translation to guide engineering efforts of rRNA processing sequences and show that divergent intergenic sequences can have significant consequences on heterologous ribosome function in E. coli. Furthermore, we identified a small subset of r-proteins that enhance the activity of refractory heterologous ribosomes with as little as 76.1% 16S rRNA sequence identity to E. coli. Finally, we find evidence that some heterologous 16S rRNAs may preferentially associate with their cognate 23S rRNAs in E. coli. Together, these results establish a quantitative and extensible method for the engineering of heterologous ribosome activity in vivo, facilitating the development of diverse ribosomes for synthetic biology applications.

## Results

**Heterologous rRNA operons complement SQ171 deficiency.** SQ171 is an E. coli strain lacking all seven chromosomal rRNA operons and carrying a single, counter-selectable plasmid bearing the wild-type rrnC operon[18,35]. To investigate the ability of heterologous rRNAs to support SQ171 cell survival, episomally encoded rRNA operons can be introduced into the strain followed by sucrose counterselection of the resident E. coli rrnC plasmid using the B. subtilis sacB cassette (Fig. 1a). Heterologous rRNA operons that yield functional heterologous ribosomes sustain SQ171 growth following sucrose counterselection. Prior work in SQ171 complementation using fully native heterologous rRNA operons has been extended to Salmonella typhimurium (96.8% 16S rRNA sequence identity to E. coli) and Proteus vulgaris rRNA (93.2%)[18].

We validated this strategy using a number of increasingly divergent heterologous rRNA operons: Salmonella enterica (97.0% 16S rRNA sequence identity to E. coli), Alteromonas macleodii (85.9%), Pseudomonas aeruginosa (85.2%), and Acinetobacter baumannii (84.3%). Heterologous rRNA derived from S. enterica robustly supported SQ171 strain growth, while rRNA derived from A. macleodii and P. aeruginosa supported growth with a minor defect (Fig. 1b). Surprisingly, we observed a substantial growth defect in SQ171 cells complemented by rRNA derived from A. baumannii despite the minor difference in sequence identity to E. coli rRNA as compared to P. aeruginosa. Motivated by these results, we extended our strategy to a total of 21 increasingly divergent rRNA operons from diverse proteobacterial species. Each of these, including an rRNA derived from the zetaproteobacteria Mariprofundus ferrooxydans (80.7%), sustained SQ171 growth (Supplementary Table 1). In line with our findings using completely native rRNA operons, chimeric heterologous-E. coli 16S rRNA fragments from gammaproteobacterial and betaproteobacterial rRNAs can often support Δ7 strain survival[17]. We further observed a linear relationship between complemented SQ171 strain fitness and 16S rRNA sequence identity, consistent with prior reports that strains relying on increasingly divergent rRNAs show comparatively reduced fitness (Fig. 1c)[17,18].

**Orthogonal translation enables quantitative heterologous ribosome assessment.** SQ171 complementation informs the capacity of a heterologous rRNA to translate the E. coli proteome of >4000 proteins[36] as well as the fulfillment of extra-catalytic roles, including integrating environmental cues to modulate translation[30] and initiating the stringent response to cellular stressors[29,37]. Additionally, we observed that SQ171 complementation pipelines require up to 5 days to observe colonies for strains relying on highly divergent rRNAs, where transformed colonies must be laboriously counter-screened due to the high escape frequency of SacB-dependent negative selection (Supplementary Fig. 1)[38,39].

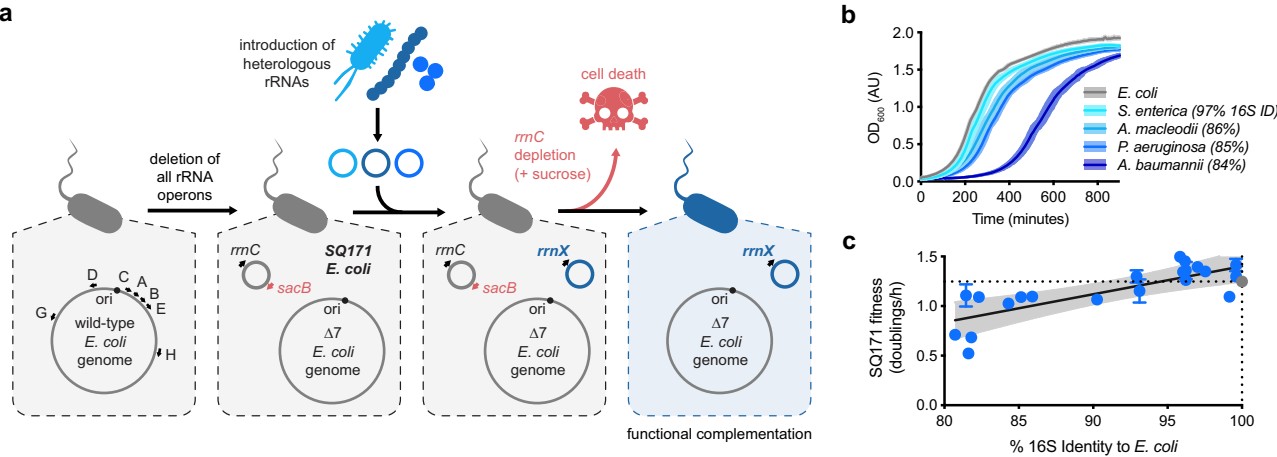

**Fig. 1 Assessment of heterologous rRNA activity via SQ171 complementation. a** Schematic representation of the SQ171 complementation assay. SQ171 *E. coli* cells lack all 7 genomic ribosomal RNA (rRNA) operons and maintain a single *rrnC* operon on a SacB counter-selectable plasmid. Introduction of a heterologous rRNA (*rrnX*) and depletion of the *E. coli rrnC* plasmid using sucrose yields cells that rely upon the heterologous ribosome for survival. **b** Growth time course of SQ171 cells bearing increasingly divergent heterologous rRNAs ($n = 2$-8). **c** Correlation between heterologous 16S rRNA sequence identity to *E. coli* (%) and SQ171 fitness (doublings/h) upon complementation (99% CI, $R^2 = 0.62$). *E. coli* rRNA control plotted in gray. Data represent the means of 1–8 biological replicates; error bars represent standard deviations for conditions with 3 or more replicates. Complete SQ171 complementation data (including precise number of replicates) are reported in Supplementary Table 1. OD optical density, AU arbitrary units, h hour. Source data are available in the Source Data File.

To overcome these technical limitations, we sought to develop an assay that delivers a single, quantifiable, translational output that is orthogonal to native ribosomal machinery and therefore independent of cell viability. We leveraged previously described orthogonal ribosome–mRNA pairs, in which the antiRBS of the 16S rRNA is engineered to exclusively translate a researcher-defined transcript bearing a complementary RBS[4,8,31–34]. This yields an orthogonal pool of ribosomes (O-ribosomes) in vivo, whose functions can be easily monitored and quantified via reporter expression (superfolder GFP; sfGFP[40]) independent of cellular survival (Fig. 2a). Importantly, wildtype ribosomes are unable to translate the orthogonal mRNA (O-mRNA) reporter, ensuring that the observed reporter activity is dependent upon engineered O-ribosomes (Fig. 2b). This orthogonal translation genetic circuit does not considerably affect cellular viability (Supplementary Fig. 2a)[4], and both O-mRNA and O-ribosome expression can be controlled via small molecule inducers to further limit the cellular burden of their production (Fig. 2a, c). We extended orthogonal translation to numerous reporter proteins, observing reporter-specific limitations on fluorescent protein functionality (Supplementary Fig. 2b–h). We accordingly identified a ten amino acid sfGFP-derived leader sequence that obviates these constraints and improves orthogonal translation for various reporters (Supplementary Fig. 3a–p).

With a robust reporter system in hand, we engineered the O-antiRBS into all 21 heterologous rRNAs capable of complementing SQ171 viability alongside an additional 13 phylogenetically more divergent rRNAs (Supplementary Table 1). We quantified the activity of all 34 O-rRNAs via orthogonal translation, finding that most rRNAs capable of supporting SQ171 growth similarly synthesized sfGFP at robust levels (Fig. 2d), with the exception of O-rRNAs derived from *Serratia marcescens* (96.0% 16S rRNA sequence identity to *E. coli*), *Vibrio cholerae* (90.3%), *P. aeruginosa* (85.2%), *A. baumannii* (84.3%), *Alcaligenes faecalis* (82.3%), *Bordetella pertussis* (81.6%), *Burkholderia cenocepacia* (81.5%), and *M. ferrooxydans* (80.7%). Notably, sfGFP translation fell markedly with phylogenetic distance from *E. coli*, wherein heterologous rRNAs exclusively derived from gammaproteobacteria and betaproteobacteria were capable of translating sfGFP

(Fig. 2d). Supporting this observation, we found a correlation between 16S rRNA sequence identity to *E. coli* and orthogonal translation activity (Fig. 2e). We similarly observed a correlation between complemented SQ171 fitness and orthogonal translation activity for each functional heterologous rRNA (Fig. 2f). Collectively, these findings support the use of orthogonal translation in lieu of SQ171 complementation to quantify the translational activity of heterologous ribosomes.

**Engineered rRNA processing improves heterologous ribosome activity.** The observed relationship between heterologous orthogonal translation activity and phylogenetic distance from *E. coli* suggested that certain elements encoded within the rRNA operon may have sufficiently diverged to restrict efficient ribosome assembly in *E. coli*. Analysis of per-base conservation[41] across the complete rRNA operons showed appreciably higher conservation scores within the ribosomal genes (16S, 23S, and 5S rRNAs) as compared to intergenic elements (Fig. 3a). Intergenic sequences flanking each rRNA gene are crucial to ribosome biogenesis as they direct pre-rRNA transcript folding and processing by RNases[3,42,43]. We hypothesized that *E. coli* RNases may fail to recognize divergent sequences on non-native rRNA transcripts, yielding immature or poorly processed heterologous ribosomes (Fig. 3b). In this case, substitution of these elements with their *E. coli* counterparts may correct the rRNA processing defect and improve overall orthogonal translation activity.

To assess the impact of putative rRNA processing on heterologous ribosome function in *E. coli*, we substituted the native intergenic sequences of each of the 34 O-rRNAs with their corresponding *E. coli* sequences (Fig. 3b). Substitution of intergenic sequences for O-rRNAs with high 16S identity to *E. coli* (96.2–99.6%) had a minimal effect on sfGFP expression (Fig. 3c). However, replacement of intergenic sequences for moderately divergent O-rRNAs (81.5–96.2%) substantially increased sfGFP expression (Fig. 3d). Interestingly, many nonfunctional O-rRNAs yielded robust sfGFP activities only after intergenic sequence replacement, namely *S. marcescens*, *V. cholerae*, *P. aeruginosa*, *A. baumannii*, and *B. cenocepacia*. Replacement of intergenic sequences for highly divergent O-rRNAs (69.8–82.3%) failed to

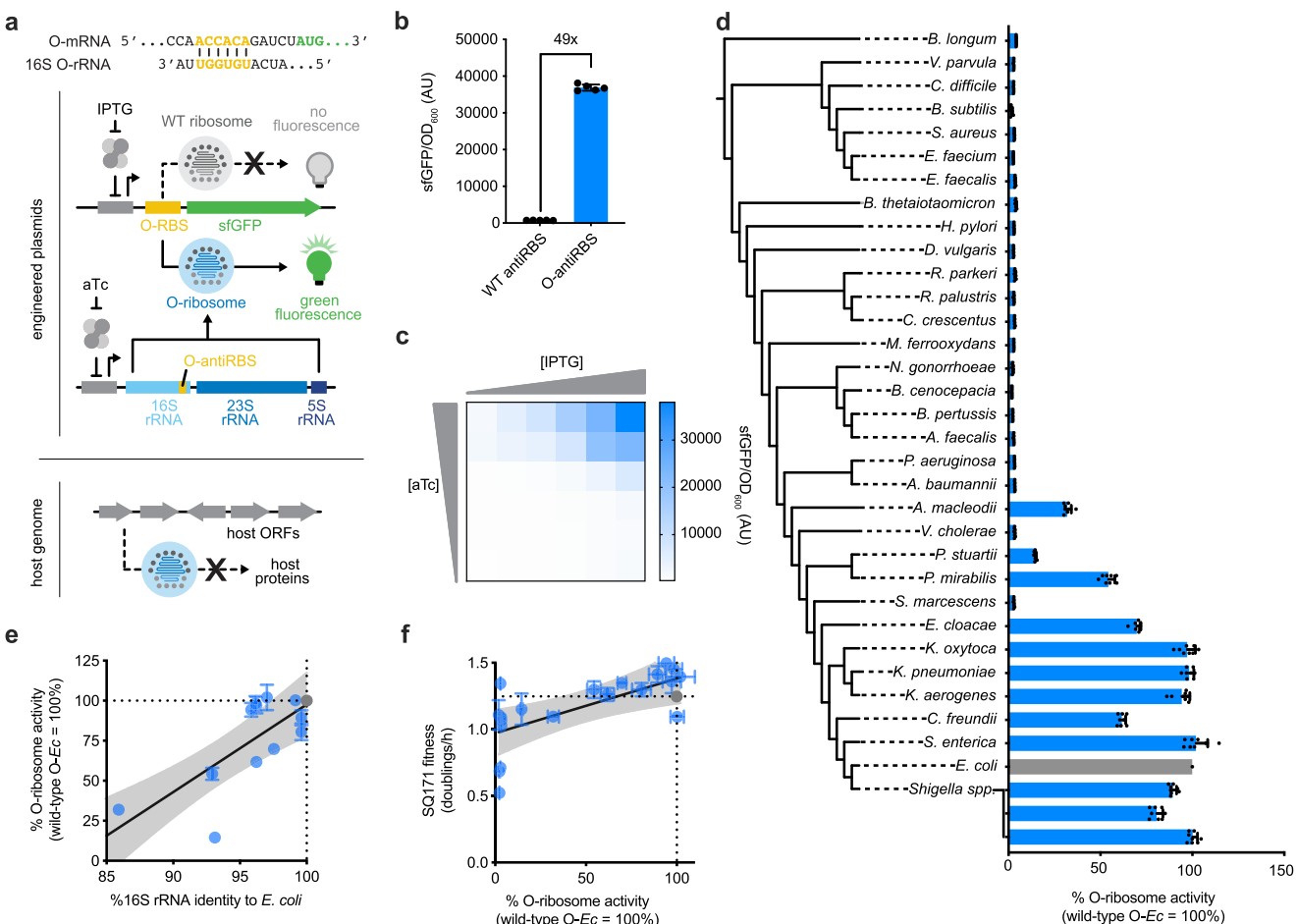

**Fig. 2 Quantification of heterologous rRNA function using orthogonal translation. a** Schematic representation of the orthogonal translation circuit. A superfolder GFP (sfGFP) reporter incorporates an orthogonal ribosome binding site (O-RBS) exclusively recognized by ribosomes that bear the complementary orthogonal anti-ribosome binding site (O-antiRBS), yielding a quantifiable fluorescent readout for heterologous ribosome activity. **b** Comparison of wild-type anti-ribosome binding site (WT antiRBS) and O-antiRBS E. coli ribosomes translating the O-RBS sfGFP reporter ($n = 5$). **c** Inducer dependence of the orthogonal translation circuit ($n = 2$). **d** Heterologous rRNA activities of 34 ribosomal RNA (rRNA) operons (blue columns) in addition to the E. coli O-rRNA control (gray column) as quantified by orthogonal translation ($n = 8$). **e** Correlation between 16S rRNA sequence identity to E. coli (%) and activity in the orthogonal translation genetic circuit (normalized to E. coli O-rRNA) for 15 functional heterologous O-rRNAs (mean activity ≥5%), illustrating a correlation between orthogonal translation activity and 16S identity (99% CI, $R^2 = 0.79$). **f** Correlation between activity in the orthogonal translation circuit and fitness in SQ171 strain complementation assays for 21 heterologous rRNAs in addition to E. coli, illustrating a linear relationship (99% CI, $R^2 = 0.47$). E. coli rRNA controls plotted in gray. Data reflect the mean and standard deviation of 1–8 biological replicates. Comprehensive SQ171 complementation and O-translation data (including precise number of replicates) reported in Supplementary Table 1. OD optical density, AU arbitrary units, wild-type O-Ec wild-type orthogonal E. coli rRNA, h hour. Source data are available in the Source Data File.

improve O-rRNA translation (Supplementary Fig. 4), suggesting that further engineering or supplementation with additional factors may be necessary to improve the activity of these highly divergent heterologous ribosomes. Finally, we reintroduced the wild-type antiRBS into the 21 engineered intergenic sequence-bearing rRNAs whose native counterparts were previously shown to support SQ171 survival, finding that SQ171 survival was maintained after intergenic sequence replacement, as was the relationship between SQ171 fitness and orthogonal translation activity (Fig. 3e). Taken together, these data suggest that rRNA processing may limit the assembly of more divergent heterologous rRNAs into functional ribosomes, and that engineering processing sites can considerably improve the activities of refractory heterologous ribosomes.

**R-protein complementation further enhances heterologous ribosome activity.** Highly divergent rRNAs (<80% 16S rRNA sequence identity to E. coli) failed to translate the orthogonal sfGFP transcript despite replacement of their intergenic

sequences, suggesting that the formation of functional heterologous ribosomes may require supplementation with additional factors. As ribosomal proteins (r-proteins) are known to co-diverge alongside their cognate rRNAs[44,45], we hypothesized that E. coli r-proteins may only be capable of binding to and forming heterologous ribosomes with rRNAs sufficiently similar to E. coli. Complementing highly divergent rRNAs with cognate r-proteins would therefore be expected to improve heterologous ribosome activity.

In prokaryotes, r-proteins are typically encoded on five operons named α, β, s10, spc, and str. R-proteins encoded within these five operons account for ~60% (12/21 SSU and 18/33 LSU in E. coli) of the full r-protein repertoire,[36] with the remaining ~40% distributed throughout the genome (Fig. 4a). Using A. baumannii O-rRNA bearing the E. coli intergenic sequences (30% activity vs. E. coli O-rRNA), we analyzed potential improvements in activity when expressing the full set of 55 cognate r-proteins distributed through seven plasmids: five corresponding to the

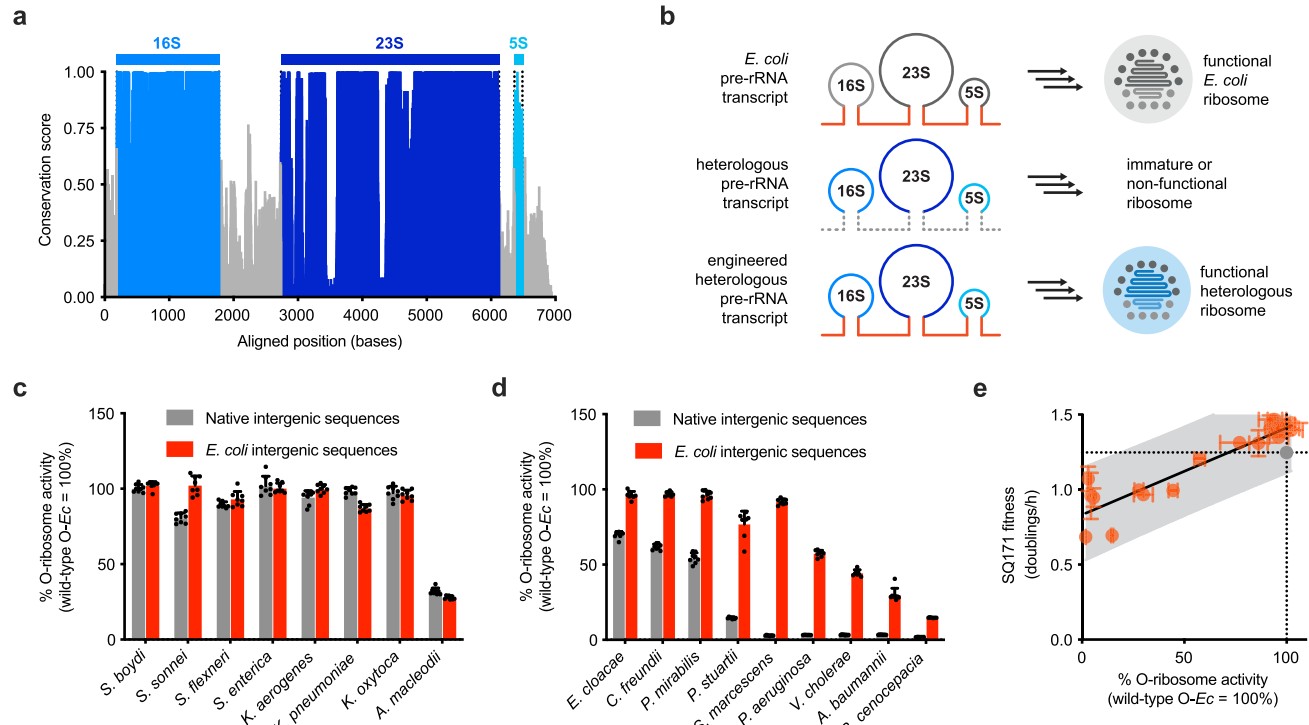

**Fig. 3 Evaluating the effects of O-rRNA intergenic sequence replacement on heterologous translation. a** Per-base sequence conservation[41] across 34 evaluated rRNA operons, showing limited conservation in intergenic regions as compared to structural rRNA genes. **b** Schematic representation of the intergenic sequence replacement strategy. **c** Effects of intergenic sequence replacement on O-rRNAs with high 16S rRNA sequence identity to *E. coli* (96.2–99.6%), as well as *A. macleodii* (85.9%), illustrating a minimal effect on orthogonal translation ($n = 8$). **d** Effects of intergenic sequence replacement on O-rRNAs with intermediate 16S rRNA sequence identity to *E. coli* (81.5–97.4%), illustrating a substantial effect on intergenic sequence replacement ($n = 8$). **e** Correlation between activity in the O-translation circuit and fitness in SQ171 strain complementation assays for 21 rRNAs evaluated after intergenic sequence replacement, showing a linear relationship. *E. coli* rRNA control plotted as a gray circle (99% CI, $R^2 = 0.84$). Data reflect the mean and standard deviation of 3–8 biological replicates. Comprehensive SQ171 complementation and O-translation data (including precise number of replicates) reported in Supplementary Table 1. Wild-type O-*Ec* wild-type orthogonal *E. coli* rRNA, O-ribosome orthogonal ribosome, h hour. Source data are available in the Source Data File.

naturally occurring r-protein operons and two artificial operons (AOs) encoding the remaining r-proteins (Fig. 4a). To capture potential epistatic interactions involving either SSU or LSU r-proteins, we enriched each artificial operon in either SSU (AO₁) or LSU (AO₂) r-proteins.

When tested alongside *A. baumannii* O-rRNA, only *Ab*AO₁ (comprising mostly SSU r-proteins) substantially improved sfGFP expression (Fig. 4b). Notably, complementation by a plasmid containing every *A. baumannii* SSU r-protein (S1-S21) yielded similar levels of activity as *Ab*AO₁, suggesting the latter contains all SSU r-proteins necessary to improve *A. baumannii* heterologous translation (Supplementary Fig. 5a). Copy-up mutations[46] to *Ab*AO₁ further improved apparent activity of this heterologous ribosome, exceeding the activity level of the *E. coli* O-rRNA (Supplementary Fig. 5b). To identify specific r-proteins responsible for this increase in heterologous ribosome activity, we sequentially deleted r-proteins from *Ab*AO₁ and found that robust sfGFP activity was maintained in all instances (Supplementary Fig. 5c), suggesting that one or more r-proteins were functionally redundant. Analysis of individual r-proteins confirmed this assessment, highlighting that expression of either *Ab*S20 or *Ab*S16 improved *A. baumannii* heterologous O-ribosome activity to levels comparable to the *E. coli* O-rRNA (Fig. 4c).

We note, however, that excessive protein overexpression alongside orthogonal translation circuits can have pleiotropic consequences on apparent translational activity. Using mCherry as a surrogate for cognate r-protein overexpression alongside

O-rRNA-dependent sfGFP production, we observe a characteristic isocost line that describes the production of two proteins under the constraints of a restricted metabolic budget (Supplementary Fig. 5d).[47] Furthermore, O-rRNA promoter choice can dramatically affect orthogonal translation activity, as promoters with repetitive elements are rapidly recombined under high expression to mitigate the associated ribosome production burden (Supplementary Fig. 5e).

We extended this analysis to *A. macleodii* O-rRNA bearing the *E. coli* intergenic sequences (17% activity vs. *E. coli* O-rRNA), finding again that only *Am*AO₁ substantially improved sfGFP expression (Fig. 4d). As this finding suggested an overlap with *A. baumannii* r-proteins that improve heterologous ribosome function, we expressed *Am*AO₁ constituent proteins alongside *Am*S20 and *Am*S16, finding that combinations of either *Am*S20 + *Am*S16 + *Am*S1 or *Am*S20 + *Am*S16 + *Am*S15 were sufficient to improve *A. macleodii* O-rRNA function to levels comparable with the *E. coli* O-rRNA (Fig. 4e). We observed a smaller but appreciable increase in apparent orthogonal translation activity using *Am*AO₂ (enriched in LSU r-proteins) (Fig. 4d, Supplementary Fig. 6a, b). Dissecting the contributions *Am*AO₂ r-proteins did not provide a comparable set of complementary r-proteins, as most genes contributed minor enhancements that collectively improved orthogonal translation activity (Supplementary Fig. 6b-e). However, these results confirmed that complementation with only a small number of cognate r-proteins can have substantial effects on heterologous ribosome function in *E. coli*.

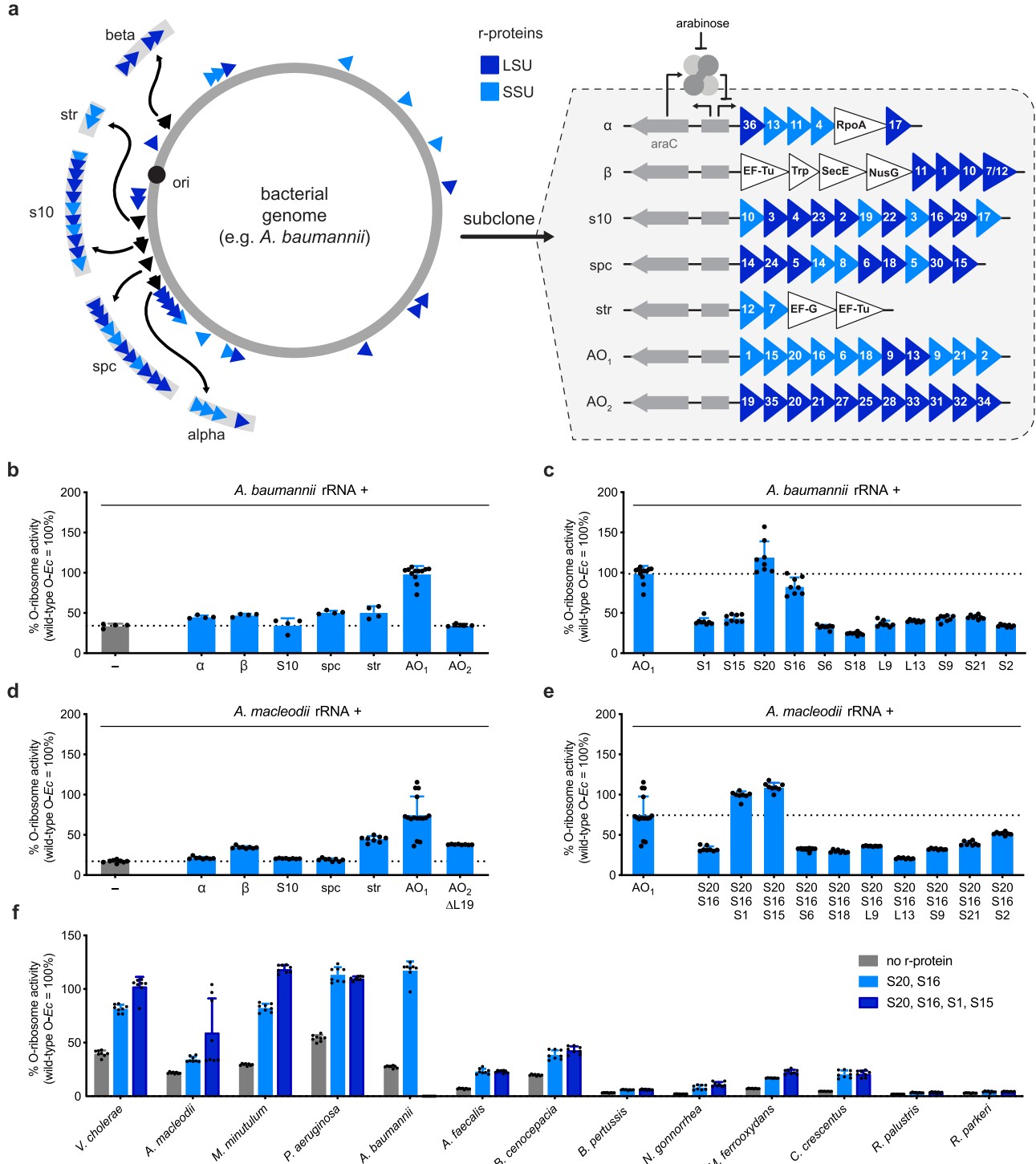

**Fig. 4 Cognate r-proteins complementation improves heterologous O-rRNA activity. a** Schematic representation of natural ribosomal protein (r-protein) genomic organization for a given microbial genome and corresponding plasmid architecture for heterologous orthogonal ribosomal RNA (O-rRNA) complementation. **b** *A. baumannii* AO$_1$ enhances cognate heterologous rRNA activity ($n = 8$ for AO$_1$; otherwise $n = 4$). **c** *A. baumannii* S20 and S16 enhance *A. baumannii* O-ribosome activity to levels comparable to *E. coli* O-ribosomes ($n = 8$ for AO$_1$; otherwise $n = 4$). **d** *A. macleodii* AO$_1$ similarly improves cognate heterologous rRNA activity ($n = 16$ for AO$_1$; otherwise $n = 8$). AO$_2$ is expressed with L19 deleted due to observed toxicity (Supplementary Figs. 6a, b). **e** Cognate S1 or S15, alongside S20 and S16, maximize *A. macleodii* O-ribosome activity ($n = 16$ for AO$_1$; otherwise $n = 8$). **f** Cognate S20, S16, S1, and S15 supplementation alongside cognate heterologous O-rRNAs. Toxicity is observed when expressing the four proteins together in *A. macleodii* and *A. baumannii* (Supplementary Fig. 7a; $n = 8$). Data reflect the mean and standard deviation of 4–16 biological replicates. Comprehensive O-translation data reported in Supplementary Table 1. Wild-type O-*Ec* wild-type orthogonal *E. coli* rRNA, SSU small subunit, LSU large subunit. Source data are available in the Source Data File.

**rRNA divergence predicts rules for cognate R-protein complementation**. The interface between rRNA and r-proteins is subject to extensive coevolution and divergence between related organisms[44,45]. Overlap between SSU r-protein complements that improved *A. baumannii* and *A. macleodii* O-rRNA activity suggested that the same r-proteins may improve the function of O-rRNAs derived from a variety of species. Indeed, the identified r-protein combinations improved activities of increasingly distant O-rRNAs: *P. aeruginosa*, *V. cholerae*, *Marinospirillum minutulum*, *A. faecalis*, *B. cenocepacia*, *Neisseria gonorrhoeae*, *M. ferrooxydans*, and *Caulobacter crescentus* (Fig. 4f).

Interestingly, this complete set of four r-proteins (S20, S16, S1, and S15) was not necessary for the observed improvement in activity for all evaluated O-rRNAs (Supplementary Fig. 7b, c). R-proteins S20 and S16 are functionally redundant when expressed alongside cognate O-rRNAs derived from species more phylogenetically related to *E. coli*: *V. cholerae*, *A. macleodii*, *M. minutulum*, *P. aeruginosa*, and *B. cenocepacia*. Uniquely, S16 has no effect on *A. faecalis* O-translation, where only S20 improves apparent activity. However, both proteins are necessary for enhanced activity when expressed alongside O-rRNAs derived from the more distant species *N. gonorrheae*, *M. ferrooxydans*, and *C. crescentus* (Supplementary Fig. 7b). Extending our analysis to the complete set of four proteins, we found that the addition of both S1 and S15 is necessary for maximal activities of *V. cholerae* and *M. minutulum* O-rRNAs, but neither r-protein has an effect when expressed alongside S20 and S16 for O-rRNAs derived from more divergent species (Supplementary Fig. 7c).

In order to determine r-proteins necessary to complement more divergent O-rRNAs, we extended our operon-based complementation approach to rRNAs derived from *B. cenocepacia* (betaproteobacteria; 81.5% 16S rRNA sequence ID to *E. coli*), *Rickettsia parkeri* (alphaproteobacteria; 76.8%), and *Enterococcus faecalis* (bacilli; 76.1%). However, the only appreciable increase in orthogonal translation was observed in *B. cenocepacia* $AO_1$, which we attributed to the contributions of S20 and S16 identified earlier (Figs. 4f and 5a–c). Having found sequence divergence from *E. coli* to be a powerful predictor of relevant features for heterologous rRNA supplementation, we manually identified 5 regions in the *E. faecalis* 16S rRNA with particularly low sequence identity to *E. coli* via pairwise alignment (Supplementary Fig. 8)[48]. As these divergent elements make extensive contacts with r-proteins in the *E. coli* ribosome (PDB: 4YBB)[49], divergence from *E. coli* in these sequences suggested an inability to efficiently bind to the requisite r-proteins.

To validate the functional relevance of these observations, we constructed variants of the *E. coli* O-rRNA in which these helices were replaced with their cognate *E. faecalis* helices, finding that orthogonal translation was abrogated in only 2 instances (transplantation of helices h9/h10 and h26; Fig. 5d). Seven r-proteins would be expected to bind these helices based on existing ribosomal structures (*Ef*S2, *Ef*S8, *Ef*S18, *Ef*S12, *Ef*S20, *Ef*S16, and *Ef*S17)[49], and indeed supplementation with this set yielded a detectable increase in orthogonal translation activity (Fig. 5e). The deletion of *Ef*S8 and *Ef*S18 from this set of proteins had no effect on activity, resulting in a set of 5 proteins that allowed *E. faecalis* O-rRNA activity to reach levels equivalent to 9.5% of the *E. coli* O-rRNA (Fig. 5e). Interestingly, this same set of 5 r-proteins was less effective than the combination of S20 and S16 for *B. cenocepacia* and *M. ferrooxydans* (81.5% and 80.1% 16S rRNA sequence identity to *E. coli*, respectively), but was more effective for the more distantly related *R. parkeri* (76.8%) and *E. faecalis* (76.1%) (Fig. 5f). We hypothesize that for O-rRNAs derived from more divergent organisms, the complete set of 5 r-proteins may be necessary to form a functional complex that cannot be formed by *E. coli* r-proteins. However, for O-rRNAs more related to *E. coli*, cognate r-proteins may compete with *E. coli* proteins for binding, forming less functional ribosomes. This finding highlights the importance of identifying the minimal subset of r-proteins necessary to improve function. Furthermore, we note that this set of 5 r-proteins is distributed across 2 naturally occurring operons in addition to the artificial operon $AO_1$, obscuring these interactions from the 7-operon approach used above. Collectively, these results indicate that rRNA/r–protein codivergence can be used to predict r-protein repertoires that enhance the activity of heterologous ribosomes in *E. coli*.

**Assessing exchange between *E. coli* and heterologous ribosome subunits**. While our analysis provided guidelines to improve refractory heterologous ribosome function, the exclusive identification of SSU r-proteins suggested that cognate heterologous LSUs may be poorly active in *E. coli*. Instead, *E. coli* LSUs may interact with heterologous SSUs to enable orthogonal translation (Supplementary Fig. 10a). Conversely, many heterologous rRNAs support SQ171 strain survival, suggesting sufficient levels of activity by many heterologous LSUs (Figs. 1c, 2f and 3e, Supplementary Table 1).

To assess the degree of association between *E. coli* LSUs and heterologous SSUs, we developed an erythromycin-dependent reporter to distinguish between genome- (*E. coli*; erythromycin-resistant) and episome-derived (heterologous; erythromycin-sensitive) LSUs. We first generated the erythromycin-resistant strain S4246, wherein all seven genomic 23S genes (*rrlA-H*) of S2060 cells were mutated (A2058U[50]) to mitigate macrolide binding in the ribosomal exit tunnel (Fig. 6a)[51,52]. Next, we introduced the ErmC leader peptide, *ermCL*, ahead of the orthogonal sfGFP reporter, ensuring that reporter translation would be abrogated via erythromycin- and *ermCL*-dependent translational stalling[53] (Fig. 6b).

To validate this sensor, we assessed the erythromycin sensitivity of an episome-derived *E. coli* orthogonal ribosome encoding or lacking the identical A2058U mutation. As a control, we leveraged a recently described stapled *E. coli* ribosome (d2d8)[7] that preferentially uses a covalently linked 23S rRNA, with or without the A2058U mutation. Using this sensor/strain combination, we find that A2058U-LSUs show no appreciable change in orthogonal translation upon erythromycin dosing (Fig. 6c). Conversely, unmutated LSUs show a marked reduction in orthogonal translation in an erythromycin dose-dependent manner. Unstapled *E. coli* LSUs lacking the A2058U mutation re-sensitized S4246 cells to erythromycin whereas the stapled counterpart did not (Fig. 6d), suggesting that plasmid-encoded LSUs may co-assemble with genome-encoded SSUs to generate erythromycin-sensitive ribosomes incapable of translating essential *E. coli* genes (Supplementary Fig. 10b).

We extended this analysis to a set of 20 functional heterologous ribosomes. For heterologous ribosomes with high 16S sequence identity to *E. coli* (≥99.2%), we observe a dramatic reduction in both sfGFP translation and cell viability (Fig. 6e, Supplementary Fig. 10c, Supplementary Table 3), reflecting appreciable exchange between host and heterologous ribosomes. Interestingly, heterologous ribosomes bearing intermediate homology (92.9–97.0%) showed extensive reduction in sfGFP signal upon erythromycin treatment with no associated viability defect. These reductions in sfGFP signal were comparable to or greater than the corresponding effect on the d2d8 stapled *E. coli* ribosome, suggesting a similar degree of association between cognate subunits. To ensure that these findings were not an artifact of heterologous ribosomes' differential antibiotic susceptibilities, we verified their erythromycin sensitivities with and without the A2058U resistance-

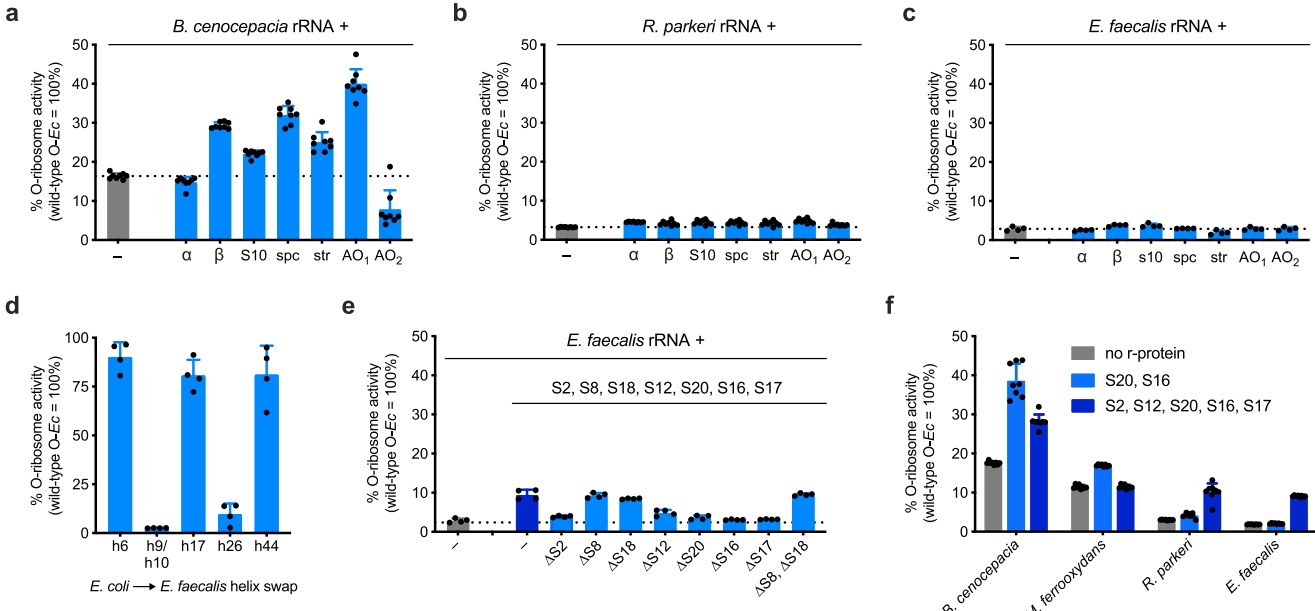

**Fig. 5 Phylogenetically guided determination of cognate r-proteins that improve highly divergent heterologous O-ribosomes in *E. coli*. a–c** No single contiguous operon substantially improves translation activity of O-rRNAs derived from **a** *B. cenocepacia* (*n* = 8), **b** *R. parkeri* (*n* = 8), or **c** *E. faecalis* (*n* = 4). **d** Only 2 of the identified 5 regions with high sequence divergence between *E. coli* and *E. faecalis* 16S rRNAs abrogate *E. coli* O-translation when replaced with cognate *E. faecalis* sequences (*n* = 4). **e** Cognate S2, S8, S18, S12, S20, S16, and S17 (which directly contact h9, h10, and h26[49]) result in an increase in *E. faecalis* O-rRNA translation. Further analysis showed that S8 and S18 are not required for this increase (*n* = 4). **f** R-proteins S2, S12, S20, S16, and S17 collectively improve cognate O-rRNA translation for highly divergent species, but are not as effective as S20 and S16 alone for less divergent species. *E. faecalis* cognate ribosomal proteins (r-proteins) are expressed from a low copy number backbone (WT RepA SC101 origin) to limit toxicity (*n* = 8). Data reflect the mean and standard deviation of 4–8 biological replicates. Comprehensive O-translation data reported in Supplementary Table 1. Wild-type O-*Ec* wild-type orthogonal *E. coli*, O-ribosome orthogonal ribosome, h helix. Source data are available in the Source Data File.

conferring mutation in SQ171 cells. For all tested heterologous ribosomes, erythromycin sensitivity was nearly identical to that of *E. coli* (Supplementary Fig. 9, Supplementary Table 2). Finally, for more divergent heterologous ribosomes (79.3–90.3%), sfGFP signal decreased only minimally upon erythromycin treatment. This signal may reflect translation using the *E. coli* LSU, where heterologous LSUs derived from highly divergent species may be poorly assembled in *E. coli*. Overall, this suggests that additional rRNA operon modifications or complementation with cognate factors may be necessary to enable the preferential usage of the heterologous LSU.

In an attempt to reduce association between heterologous and *E. coli* ribosomal subunits, we extended the above rRNA stapling approach[7] to the same 20 heterologous ribosomes. This staple appears to be extensible to rRNAs with high sequence identity (≥99.2% 16S) to *E. coli* (Supplementary Fig. 10d-f). However, for most rRNAs, this approach does not increase erythromycin sensitivity. Hypothesizing that the d2d8 linkers were not suited to heterologous LSUs, we therefore generated "hybrid" ribosomes comprising heterologous SSUs stapled to *E. coli* LSUs. Finding that these hybrid ribosomes vary considerably in erythromycin sensitivity, we postulate that the rRNA linker would require independent optimization for each heterologous ribosome. Collectively, these data suggest that intermediately divergent heterologous SSUs may preferentially associate with the cognate LSUs in *E. coli*, and could serve as starting points for future engineering efforts.

## Discussion

We constructed a library of 34 heterologous ribosomes derived from species across a broad phylogenetic range and expressed in *E. coli*. We assayed the functionality of these ribosomes using both Δ7 strain complementation and orthogonal translation, and

found a high degree of correlation between the two assays. Replacement of intergenic sequences with those of *E. coli*, as well as supplementation with only a small subset of r-proteins (S20, S16, S1, and S15), improved expression from orthogonal heterologous rRNAs. While O-rRNAs with high sequence identity (as little as 96.2% 16S rRNA sequence identity to *E. coli*) natively translated sfGFP at robust levels, substitution of intergenic sequences allowed for O-rRNAs as divergent as *P. mirabilis* (92.9%) to translate the orthogonal transcript at levels similar to the *E. coli* O-rRNA. Supplementation with r-proteins S20 and S16 allowed for similarly robust levels of translation from O-rRNA derived from *A. baumannii* (84.3%). Using a more extensive set of r-proteins, we were able to observe heterologous translation from O-rRNAs as diverged as *E. faecalis* (76.1%). Finally, using an erythromycin-dependent reporter, we found that a subset of heterologous SSUs appeared to preferentially associate with their cognate LSUs.

Collectively, our findings establish orthogonal translation as a viable alternative to Δ7 complementation for evaluating the function of heterologous rRNAs and suggest generalizable strategies for enhancing heterologous rRNA function. Interestingly, of the four r-proteins found to be broadly important for O-rRNA function, only two (S1 and S16) have been found to be essential for viability in *E. coli* via gene knockout,[54,55] suggesting that essentiality cannot serve as a predictor of crucial factors enhancing O-rRNA function. We therefore sought to determine whether we could derive rules for predicting r-proteins necessary for complementing heterologous O-rRNAs. Using sequence similarities of heterologous SSU r-proteins to their *E. coli* homologues, we found that r-proteins empirically discovered to be crucial for complementation (S20, S16, and S15) tend to be those with the lowest sequence similarity (Supplementary Fig. 11a-n). Particularly in the case of S20 and S16, this may reflect their roles as primary binders to the rRNA[56], which are

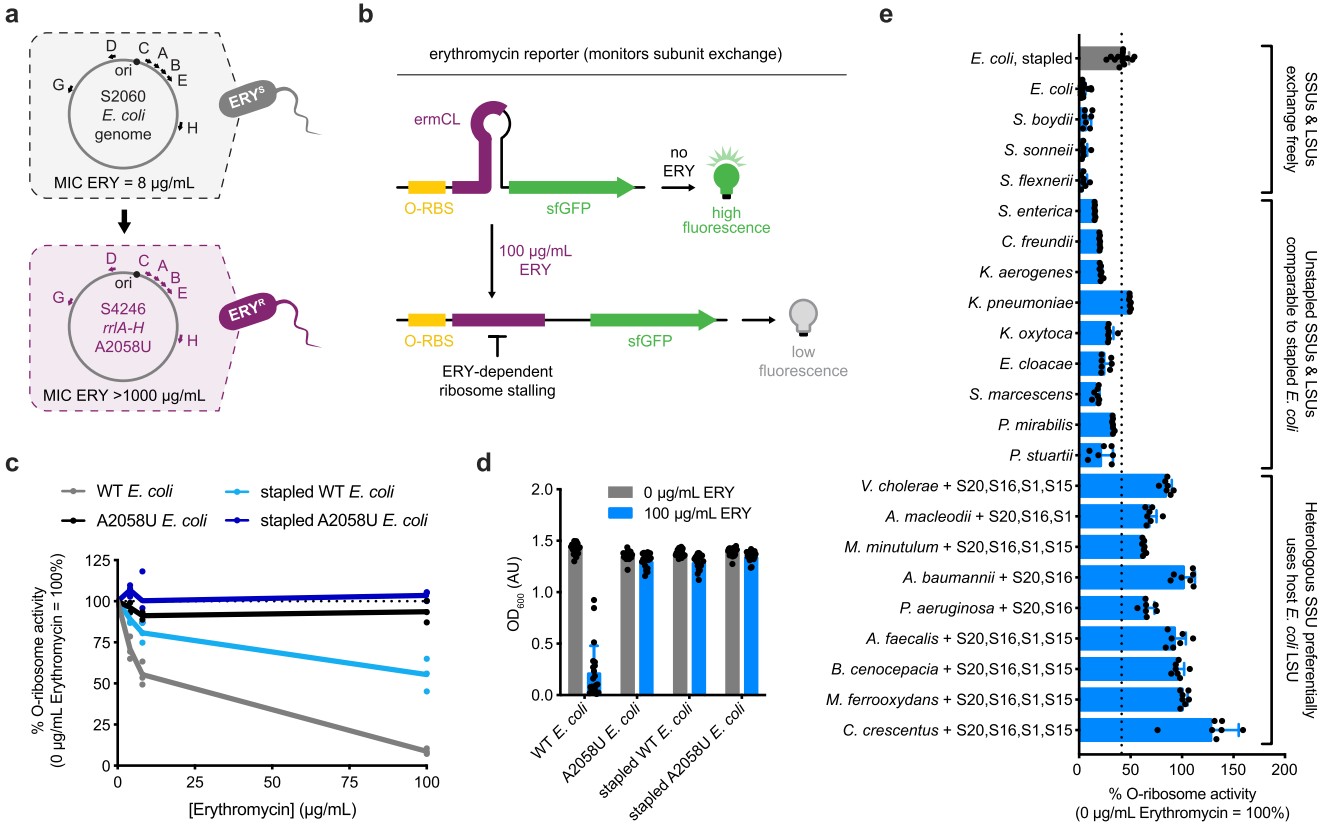

**Fig. 6 The erythromycin-dependent orthogonal translation reporter system. a** Creation of the erythromycin (ERY)-resistant *E. coli* strain S4246. All 7 *rrl* (A-H) 23S ribosomal RNA (rRNA) genes were mutated (A2058U) via oligonucleotide recombineering to endow high erythromycin resistance (ERY; MIC > 1000 µg mL$^{-1}$). **b** Schematic representation of the ERY-dependent superfolder GFP (sfGFP) reporter. In the absence of ERY, sfGFP is efficiently translated via orthogonal translation. Addition of ERY (100 µg mL$^{-1}$) promotes translation stalling at the ermC leader peptide (ermCL), abrogating sfGFP translation by ERY-sensitive large subunits (LSUs). **c** Free and stapled ERY-sensitive LSUs show a marked reduction in sfGFP production at high inhibitor concentrations, whereas the corresponding ERY-resistant (23S A2058U) LSUs show no appreciable change in activity ($n = 2$ for A2058U *E. coli* at 4 µg/mL ERY; otherwise $n = 3$). **d** ERY-sensitive LSUs re-establish strain sensitivity to ERY due to free subunit exchange between episomally and genomically derived ribosomes ($n = 21$). **e** Evaluation of intersubunit exchange using the ERY-dependent reporter system. Heterologous ribosomes with high 16S sequence identity to *E. coli* (≥99.2%) appear to freely exchange with host subunits, while heterologous ribosomes with intermediate sequence identity (97.0–92.9%) preferentially associate with cognate subunits at a rate comparable to the stapled *E. coli* ribosome. More divergent heterologous ribosomes (90.3–79.3%) preferentially utilize *E. coli* large subunits ($n = 28$ for *E. coli* and $n = 14$ for *E. coli*, stapled; otherwise $n = 7$). Data for each ribosome is normalized to its corresponding sfGFP signal at 0 µg mL$^{-1}$ ERY. Data reflect the mean and standard deviation of the indicated biological replicates. Comprehensive data reported in Supplementary Table 3. MIC = minimum inhibitory concentration; OD optical density, O-RBS orthogonal RBS, O-ribosome orthogonal ribosome, SSU small subunit. Source data are available in the Source Data File.

known to diverge more rapidly.[44,45] All three proteins, however, play important roles in the earliest stages of 30S assembly.[56] Similarly, S1 often interacts with mRNAs in proximity to the RBS during translational initiation,[57,58] suggesting that it may play a role in mediating correct RBS/antiRBS interactions using noncanonical (orthogonal) pairs.

In the future, it may be possible that heterologous rRNAs, specifically those of relevance to human health, can be expressed in *E. coli* for the high-throughput discovery of ribosome-targeting antibiotics. More broadly, heterologous ribosomes and strategies described here may serve as alternative starting points for the discovery and laboratory evolution of novel translational properties. Through the use of such ribosomes, synthetic biologists can take advantage of the myriad functionalities that natural Darwinian evolution has generated in phylogenetically diverse bacterial ribosomes.

## Methods

**General methods**. Unless otherwise noted, all PCRs were performed using Phusion U HotStart DNA Polymerase (Life Technologies). Water was purified using a MilliQ water purification system (MilliporeSigma). Antibiotics (Gold Biotechnology) were used at the following concentrations for plasmid selection, unless otherwise noted: 30 µg mL$^{-1}$ kanamycin, 40 µg mL$^{-1}$ chloramphenicol, 50 µg mL$^{-1}$ carbenicillin, 100 µg mL$^{-1}$ spectinomycin. Antibiotics were used at one-third concentrations for strains bearing three unique plasmids. Unless otherwise noted, all DNA manipulations were performed in NEB Turbo cells (New England Biolabs) or Mach1F cells (Mach1 T1$^R$ cells (Thermo Fisher Scientific) mated with F' episome of the previously described S2060[59] strain). All fluorescence and luminescence assays were carried out using *E. coli* S2060[59].

**Chemically competent cell preparation**. Chemically competent cells were used for all cloning and assay pipelines. A glycerol stock of the appropriate strain was used to start a 2 mL culture supplemented with the appropriate antibiotics and grown up overnight at 30 °C at 300 RPM. The saturated culture was diluted 1:1000 in 50 mL 2xYT (United States Biological) with appropriate antibiotics and grown to OD$_{600}$ = 0.3–0.5 in a 37 °C shaker at 300 RPM. Cells were pelleted in a pre-chilled conical tube (VWR) by centrifugation at 8000 $g$ for 10 min at 4 °C. The supernatant was removed and the cells were resuspended in approximately 20 mL 10% glycerol, then pelleted by centrifugation at 8000 $g$ for 10 min at 4 °C. The supernatant was removed and the cells were resuspended in chilled TSS buffer (2xYT media supplemented with 5% DMSO, 10% PEG2250, 2 mM MgCl2). Cells were flash frozen in liquid N$_2$ at 100 µL aliquots and transferred to −80 °C storage.

**USER cloning**. Plasmids were constructed using USER cloning, or a combination of USER cloning and overlap extension PCR. In USER cloning, primers are

designed to include a deoxyuracil base approximately 10–20 bases from the 5′ end of the primer; the region between the deoxyuracil base and the 5′ end of the primer is known as the "USER junction" and specifies the homology necessary for plasmid assembly. USER junctions were designed to have a 42 °C < $T_m$ < 70 °C, minimal secondary structure, and begin with a dA and end with a dT (the latter is replaced with a dU to act as a substrate for uracil DNA glycosylase during assembly). PCR products were gel purified using QIAquick Gel Extraction kit (Qiagen) and eluted to a final volume of 10 µL. Fragments were quantified using a NanoDrop 1000 Spectrophotometer (Thermo Fisher Scientific). For assembly, PCR products with complementary USER junctions were added in an equimolar ratio (0.1–1 pmol each) in a 10 µL reaction containing 0.75 units DpnI (New England Biolabs), 0.75 units USER (Uracil-Specific Excision Reagent; Endonuclease VIII and Uracil-DNA Glycosylase) enzyme (New England Biolabs), 1 unit of CutSmart Buffer (50 mM potassium acetate, 20 mM Tris-acetate, 10 mM magnesium acetate, 100 µg mL$^{-1}$ BSA at pH 7.9; New England Biolabs). Reactions were incubated at 37 °C for 20 min, then heated at 80 °C and slowly cooled to 12 °C at 0.1 °C/s in a thermocycler. Inserts of plasmids "AO₁", "AO₂", and "S1-S21," consisting of many small fragments, were cloned using overlap extension PCR. Primers were designed containing ~15 bp overhangs complementary to the adjoining fragment. Individual fragments were amplified and gel purified as above, then 0.2 picomoles of each fragment was used in a 200 µL PCR reaction to join each fragment together. This fragment was gel purified and USER assembly was used for cloning into the appropriate plasmid backbone. Assembled constructs were heat-shocked into chemically competent NEB Turbo or Mach1F cells: 100 µL 2x KCM (100 mM KCl, 30 mM CaCl₂, 50 mM MgCl₂ in MilliQ H₂O) was added to 100 µL cells alongside plasmid DNA. Cells were incubated on ice for 15 min, heat shocked at 42 °C for 2 min, and placed back on ice for 2 min. Cells were recovered in 1 mL 2xYT media at 37 °C with shaking at 300 RPM for a minimum of 45 min. Cells were streaked on 1.8% agar-2xYT plates supplemented with the appropriate antibiotic(s).

**Amplification of ribosomal operons and R-proteins.** Ribosomal DNA, contiguous r-protein operons, and single r-protein ORFs were amplified from bacterial strains or the corresponding gDNA. Direct amplification from bacterial strains required boiling at 95 °C for 10 min in MilliQ water prior to PCR for efficient amplification. In cases where a non-type strain was used, universal primers (Supplementary Table 4) were used to amplify a ~900 bp fragment from the bacterial genome to include a partial 16S element for subcloning and sequencing, allowing for closest sequenced genome determination. For species with high sequence variability between ribosomal operons, a representative operon was chosen based on maximal sequence homology to *E. coli*.

**Bacterial strain genomic modifications.** The erythromycin-resistant strain S4246 was generated using conventional recombineering[51,52]. Briefly, chemically competent S2060 cells were transformed with pKD46[52] and plated on 2xYT agar plates supplemented with 50 µg mL$^{-1}$ carbencillin at 30 °C. A single colony was picked, grown at 30 °C in 2xYT liquid medium supplemented with 50 µg mL$^{-1}$ carbencillin and 10 mM arabinose, and made chemically competent when the culture reached the appropriate OD₆₀₀. Chemically competent S2060/pKD46 cells were transformed with the phosphothiorated recombineering oligonucleotide AB5708 (Supplementary Table 4) to introduce the *rrlA-H* A2058U mutation on replichore 2[51]. Following recovery for 3 h at 30 °C, transformed cells were plated on 2xYT agar plates supplemented with 1000 µg mL$^{-1}$ erythromycin and incubated at 37 °C to cure the resident pKD46 plasmid. Following overnight growth, single colonies were picked into 2xYT liquid medium supplemented with 50 µg mL$^{-1}$ streptomycin, 10 µg mL$^{-1}$ tetracycline, and 1000 µg mL$^{-1}$ erythromycin and allowed to grow overnight at 37 °C. To assess the degree of *rrlA-H* mutagenesis, cultures were used as PCR templates using primers AB5710 and AB5711 (Supplementary Table 4), and the PCR products were treated with the endonuclease HpyCH4III (New England Biolabs) according to the manufacturer's guidelines. Wild-type *rrlA-H* genotypes show no digestion under these conditions, whereas complete conversion results in complete PCR product digestion. Intermediate (incomplete) digestion indicated in incomplete conversion of all seven genomic alleles. The completely converted strain S4246 was confirmed to be sensitive to the following antibiotics (ensures no resistance crosstalk with plasmid-borne markers): carbenicillin (50 µg mL$^{-1}$), spectinomycin (100 µg mL$^{-1}$), chloramphenicol (40 µg mL$^{-1}$), and kanamycin (30 µg mL$^{-1}$). The strain was confirmed to be resistant to the following antibiotics: streptomycin (50 µg mL$^{-1}$), tetracycline (10 µg mL$^{-1}$), and erythromcyin (1000 µg mL$^{-1}$).

**Fluorescence assays.** For orthogonal translation assays, S2060 chemically competent cells were transformed with the *E. coli* O-rRNA plasmid and the relevant orthogonal reporter plasmid. Transformants were streaked on 1.8% agar-2xYT plates supplemented with kanamycin and carbenicillin. Plates were grown in a 37 °C incubator for 16 h. Colonies were picked into 500 µL DRM (United States Biological)[60] (supplemented with kanamycin, carbenicillin, 1 mM IPTG +/- 1000 ng mL$^{-1}$ aTc) in deepwell plates (VWR) and grown at 37° C with shaking at 900 RPM for 20 hours. To assay heterologous O-rRNA function, chemically competent cells carrying the sfGFP reporter plasmid were prepared (S2060.sfGFP) and transformed with the appropriate O-rRNA plasmid. *E. coli* O-rRNA was always transformed alongside

experimental O-rRNAs as a positive control. Transformants were streaked out and picked into media as above.

To assay r-protein effects on heterologous O-rRNA function, S2060.sfGFP chemically competent cells were co-transformed with the appropriate O-rRNA plasmid and r-protein plasmid. As a positive control, *E. coli* O-rRNA was transformed alongside an mCherry expression plasmid. In the absence of r-protein supplementation, heterologous O-rRNAs were transformed with mCherry to maintain consistent growth rates and antibiotic selection markers. Transformants were streaked on 1.8% agar-2xYT plates supplemented with kanamycin, carbenicillin, chloramphenicol, and 200 mM glucose, picked into DRM supplemented with kanamycin, carbenicillin, chloramphenicol, 1 mM IPTG, 1000 ng mL$^{-1}$ aTc, +/- 10 mM arabinose, and grown up as above. For assays using the erythromycin-dependent reporter, DRM was supplemented with 100 µg mL$^{-1}$ erythromycin in addition to the above inducers and antibiotics.

To quantify fluorescence output, 150 µL of each culture were aliquoted into a 96-well black wall, clear bottom plate (Costar). OD₆₀₀ and the appropriate excitation and emission wavelengths were used for fluorescence measurements (Supplementary Table 5) using either a SpectraMax M3 (Molecular Devices) or Spark (Tecan) plate reader running SoftMax Pro v6.4 or SparkControl v2.3, respectively. Fluorescence was normalized to OD₆₀₀ after blank media subtraction. Data were normalized to *E. coli* O-rRNA sfGFP/OD₆₀₀ and expressed as a percentage. When assaying the effects of r-protein complementation, data were normalized to *E. coli* O-rRNA (sfGFP signal) upon mCherry control plasmid induction.

**SQ171 cell viability assay.** Chemically competent SQ171[18,35] cells were transformed with heterologous rRNAs as described above and recovered for up to 7 h in 2xYT in a 37 °C shaker. The recovery culture was centrifuged at 10,000 RCF for 2 min, then the pellet was resuspended in 100 µL MilliQ water. The resuspended cells were diluted serially in seven, 10-fold increments to yield eight total samples (undiluted, 10$^1$-, 10$^2$-, 10$^3$-, 10$^4$-, 10$^5$-, 10$^6$-, and 10$^7$-fold diluted). To determine the efficiencies of EP transformation and counter-selectable plasmid curing, 3 µl of diluted cells were plated on 1.8% agar-2xYT plates (United States Biological) supplemented with spectinomycin (100 µg mL$^{-1}$) and carbenicillin (50 µg mL$^{-1}$), with or without 5% sucrose (Millipore Sigma). For picking single colonies, the remaining undiluted cells were plated on 1.8% agar-2xYT plates (United States Biological) containing spectinomycin (100 µg mL$^{-1}$), carbenicillin (50 µg mL$^{-1}$), and 5% sucrose. All plates were grown for 16–120 h in a 37 °C incubator.

For erythromycin sensitivity assays, colonies transformed with the appropriate rRNA EP and surviving sucrose selection were picked and grown in DRM containing spectinomycin (100 µg mL$^{-1}$), carbenicillin (50 µg mL$^{-1}$), and 5% sucrose, with or without kanamycin (30 µg mL$^{-1}$). Following overnight growth, cultures were diluted 100-fold into fresh DRM containing spectinomycin (100 µg mL$^{-1}$) and carbenicillin (50 µg mL$^{-1}$). Wild-type strains were evaluated at 12 erythromycin concentrations from 1000 – 0.002 µg mL$^{-1}$ in 3-fold dilution increments, and A2058U-23S strains were evaluated from 2000 – 1 µg mL$^{-1}$ in two-fold dilution increments. Colonies that survived selection in kanamycin were excluded from final analysis, as survival in kanamycin indicates persistence of the resident pCSacB plasmid (which carries a KanR resistance cassette). Following overnight growth, 200 µL of each culture were aliquoted into a 96-well black wall, clear bottom plate (Costar). OD₆₀₀ was measured using a Spark (Tecan) plate reader. IC₅₀ curves were fit using GraphPad Prism version 8.

For all other SQ171 assays, colonies transformed with the appropriate EP and surviving sucrose selection were picked and grown in DRM containing spectinomycin (100 µg mL$^{-1}$), carbenicillin (50 µg mL$^{-1}$), and 5% sucrose. Following growth of the EP-carrying strains for up to 3 days, cultures were glycerol stocked. Overnight cultures were started from these glycerol stocks in DRM containing spectinomycin (100 µg mL$^{-1}$), carbenicillin (50 µg mL$^{-1}$), and 5% sucrose. Following overnight growth, cultures were diluted 100-fold into fresh DRM containing spectinomycin (100 µg mL$^{-1}$) and carbenicillin (50 µg mL$^{-1}$). From the diluted cultures, 200 µL of each culture were transferred to a 96-well black wall, clear bottom plate (Costar), topped with 20 µL of mineral oil, and the OD₆₀₀ was measured every 5 min over 15 h. Separately, 400 µL of each diluted culture were supplemented with kanamycin (30 µg mL$^{-1}$) and grown in deepwell plates (VWR) at 37 °C with shaking at 900 RPM. Colonies that survived selection in kanamycin were excluded from final analysis, as survival in kanamycin indicates persistence of the resident pCSacB plasmid (which carries a KanR resistance cassette). The doubling time of each culture was calculated using the Growthcurver package (version 0.3.0)[61] in *R* (version 3.5.2). Data for SQ171 assay complementation in the main text is reported as fitness in doublings/h, obtained by taking the inverse of doubling time in minutes and multiplying by 60.

**Phylogenetic analyses.** To calculate sequence identities, 16S sequences of all rRNAs used in the study were aligned using Clustal Omega with default settings[48]. The phylogenetic tree (Fig. 2d) was constructed using phylogenetic relationships derived from the Genome Taxonomy database (GTDB)[62]. In short, the entire bacterial GTDB phylogenetic tree (release 86.1) was downloaded from https://data.ace.uq.edu.au/public/gtdb/data/releases/release86/86.1/. The phylogenetic tree was pruned to include only species of interest (see Supplementary Table 6 for the correspondence between species names and respective GTDB representative

genomes) using the *Ape* package (version 5.3) in *R* (version 3.5.2). The pairwise distances between the tips in the pruned trees were computed using the *Ape* package[63]. The tree was visualized using iTOL (version 5.7)[64]. Per-base conservation[41] (Fig. 3a) was calculated using the Biostrings (version 2.52.0) (https://bioconductor.org/packages/Biostrings) and TFBSTools[65] (version 1.20.0) packages in *R* (version 3.5.2).

**Protein sequence similarity analysis**. To analyze sequence similarities of r-proteins, RefSeq proteomes of relevant species were downloaded and a local BLAST database was created from these proteomes (using BLAST version 2.7.1). *E. coli* SSU r-protein sequences were queried against the database using local *blastp* with default parameters using the BLOSUM62 similarity matrix. Hits were filtered to those annotated with "30S," "SSU," or "ribosomal protein."

**Reporting summary**. Further information on research design is available in the Nature Research Reporting Summary linked to this article.

## Data availability
The authors declare that all the data supporting the findings of this study are available within the paper and its supplementary information files. The source data for Figs. 1b-c, 2b-f, 3a, 3c-e, 4b-f, 5a-f, 1c-e, and Supplementary Figs. 2a-h, 3b-p, 4, 5a-e, 6a-e, 7a-c, 9a-j, 10a, 10c-f, and 11a-n are provided as a Source Data File. The Genome Taxonomy Data Base phylogenetic tree used for phylogenetic analysis can be downloaded at https://data. ace.uq.edu.au/public/gtdb/data/releases/release86/86.1/. See Supplementary Table 6 for the correspondence between species names, NCBI taxIDs, NCBI species taxIDs, NCBI strain identifiers, and respective GTDB representative genomes. Key plasmids used in this study have been deposited in Addgene (see Supplementary Table 8 for Addgene IDs). All other relevant data are available from the authors upon reasonable request. Source data are provided with this paper.

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

## Acknowledgements

The authors would like to thank all members of the Badran lab for helpful discussions. This work was supported by the National Institutes of Health Director's Early Independence Award (DP5-OD-024590), the NASA Exobiology Program (NNH17ZDA001N-EXO), and the Broad Institute of MIT and Harvard. The authors thank the following individuals for providing templates for rDNA amplification: Arnaud Guttierez, Bernardo Cervantez, Jim Gomez, Roby Bhattacharyya, Daria van Tyne, Austin Cole, Aaron James Dy, Allison Coe, Ning Mao, Nick Lyons, Becky Lamason, Kevin Gozzi, Vayu Maini Rekdal, Samantha Palace, Daniel Jun, and Jacob Cohen. The authors additionally thank the laboratories of Deborah Hung, Paul Blainey, and James Collins for equipment use and assistance. The authors are grateful to Erik Carlson and Mike Jewett for sharing materials used in this work.

## Author contributions

N.S.K. designed the research, performed experiments, and analyzed data. R.F. designed, carried out, and analyzed SQ171 and sfGFP assays. S.B. performed rRNA and r-protein phylogenetic analyses. G.D.C. generated and analyzed fluorescent protein reporters. A.H. B. conceived of and designed the research, performed experiments, analyzed data, and supervised the research. N.S.K. and A.H.B. wrote the manuscript, with contributions from all authors.

## Competing interests

N.S.K. and A.H.B. declare competing interests. The Broad Institute has filed a patent application directed to work described in this article; N.S.K. and A.H.B. are inventors on that application. The provisional applications were filed in the United States, entitled "Heterologous Ribosome Generation, Assessment and Compositions Thereof". The associated PCT/US2020/041905, was filed on the 14th of July 2020. This patent covers methods for generating heterologous ribosomes and assessing/improving their activity. R.F., S.B., and G.D.C. declare no competing interests.
