## [Peer Review File · Nature Communications]

Reviewers' Comments:

Reviewer #1:

Remarks to the Author:

In this re-submitted manuscript Kolber and colleagues further refine their study examining the expression of rRNA operons from diverse bacteria in *E. coli*. They improve the text of the manuscript and present new data to address comments from the previous rounds of review. Although ultimately fruitless, I appreciate the authors' attempts to biochemically determine if association between the heterologous o-SSUs and native LSUs can occur. This highlights the challenges in engineering such a critical hub for cellular activities. Despite this shortcoming I believe the manuscript is now suitable for publication in *Nature Communications*. This conclusion is based on the substantial new, interesting data presented on the activities of diverse rRNAs in *E. coli*, the contributions of RNA processing and heterologous ribosomal proteins to their activity, and the new methodological advances outlined in the manuscript, including improved O-mRNA reporters. Furthermore, adjustments to the text now clearly highlight the potential pitfalls as well as the advantages for these systems. I look forward to seeing the authors' work online.